# Genetic Prognostic Factors in Adult Diffuse Gliomas: A 10-Year Experience at a Single Institution

**DOI:** 10.3390/cancers16112121

**Published:** 2024-06-01

**Authors:** Amir Barzegar Behrooz, Hadi Darzi Ramandi, Hamid Latifi-Navid, Payam Peymani, Rahil Tarharoudi, Nasrin Momeni, Mohammad Mehdi Sabaghpour Azarian, Sherif Eltonsy, Ahmad Pour-Rashidi, Saeid Ghavami

**Affiliations:** 1Department of Human Anatomy and Cell Science, University of Manitoba College of Medicine, Winnipeg, MB R3E 0J9, Canada; am.barzegar.behrooz@gmail.com; 2Electrophysiology Research Center, Neuroscience Institute, Tehran University of Medical Sciences, Tehran 1416634793, Iran; hlatifin@gmail.com; 3Brain Cancer Research Group, Department of Cancer, Asu Vanda Gene Industrial Research Company, Tehran 1533666398, Iran; rahil_tarharoudi@yahoo.com (R.T.); nasrinmomeni96@gmail.com (N.M.); 4Department of Plant Production and Genetics, Bu-Ali Sina University, Hamedan 6517838623, Iran; h.ramandi@basu.ac.ir; 5Department of Molecular Physiology, Agricultural Biotechnology Research Institute of Iran, Agricultural Research Education and Extension Organization (AREEO), Karaj 7155863511, Iran; 6Department of Biostatistics, Asu Vanda Gene Industrial Research Company, Tehran 1533666398, Iran; 7Department of Molecular Medicine, National Institute of Genetic Engineering and Biotechnology, P.O. Box 14965/161, Tehran 1497716316, Iran; 8School of Biological Sciences, Institute for Research in Fundamental Sciences (IPM), Tehran 1953833511, Iran; 9College of Pharmacy, Rady Faculty of Health Sciences, University of Manitoba, Winnipeg, MB R3T 2N2, Canada; payam.peymani@umanitoba.ca (P.P.); sherif.eltonsy@umanitoba.ca (S.E.); 10Department of Molecular and Cellular Sciences, Faculty of Advanced Science and Technology, Tehran Medical Sciences, Islamic Azad University, Tehran 1477893855, Iran; 11Department of Biotechnology, Asu Vanda Gene Industrial Research Company, Tehran 1533666398, Iran; moh.sabaghpour@gmail.com; 12Department of Neurosurgery, Sina Hospital, Tehran University of Medical Sciences, Tehran 1416634793, Iran; 13Research Institute of Oncology and Hematology, Cancer Care Manitoba-University of Manitoba, Winnipeg, MB R3E 0V9, Canada; 14Biology of Breathing Theme, Children Hospital Research Institute of Manitoba, University of Manitoba, Winnipeg, MB R3T 2N2, Canada

**Keywords:** glioma, glioblastoma, overall survival, genomic profiling, tumor heterogeneity, tumor location

## Abstract

**Simple Summary:**

Despite recent advances in diagnosing and treating glioma, the prognosis remains poor. It is crucial to investigate its clinical characteristics and prognostic factors to provide a basis for treating and managing patients with glioma. We examined how clinical variables and molecular profiles may have affected overall survival (OS) over the past 10 years. A correlation between IDH status, TERT mutations, MGMT methylation, and tumor location characteristics was also studied. Our study demonstrates that patients with higher tumor histological grades who had received adjuvant radiotherapy exhibited isocitrate dehydrogenase 1 (IDH1) mutations or presented with wildtype telomerase reverse transcriptase promoter (TERTp) experienced improved OS.

**Abstract:**

Gliomas are primary brain lesions involving cerebral structures without well-defined boundaries and constitute the most prevalent central nervous system (CNS) neoplasms. Among gliomas, glioblastoma (GB) is a glioma of the highest grade and is associated with a grim prognosis. We examined how clinical variables and molecular profiles may have affected overall survival (OS) over the past ten years. A retrospective study was conducted at Sina Hospital in Tehran, Iran and examined patients with confirmed glioma diagnoses between 2012 and 2020. We evaluated the correlation between OS in GB patients and sociodemographic as well as clinical factors and molecular profiling based on IDH1, O-6-Methylguanine-DNA Methyltransferase (MGMT), TERTp, and epidermal growth factor receptor (EGFR) amplification (EGFR-amp) status. Kaplan–Meier and multivariate Cox regression models were used to assess patient survival. A total of 178 patients were enrolled in the study. The median OS was 20 months, with a 2-year survival rate of 61.0%. Among the 127 patients with available IDH measurements, 100 (78.7%) exhibited mutated IDH1 (IDH1-mut) tumors. Of the 127 patients with assessed MGMT promoter methylation (MGMTp-met), 89 (70.1%) had MGMT methylated tumors. Mutant TERTp (TERTp-mut) was detected in 20 out of 127 cases (15.7%), while wildtype TERTp (wildtype TERTp-wt) was observed in 107 cases (84.3%). Analyses using multivariable models revealed that age at histological grade (*p* < 0.0001), adjuvant radiotherapy (*p* < 0.018), IDH1 status (*p* < 0.043), and TERT-p status (*p* < 0.014) were independently associated with OS. Our study demonstrates that patients with higher tumor histological grades who had received adjuvant radiotherapy exhibited IDH1-mut or presented with TERTp-wt experienced improved OS. Besides, an interesting finding showed an association between methylation of MGMTp and TERTp status with tumor location.

## 1. Introduction

Gliomas are primary brain lesions involving cerebral structures without well-defined boundaries and constitute the most prevalent central nervous system (CNS) neoplasms. Glioblastoma (GB) is a glioma of the highest grade with a very dismal prognosis. Gliomas can develop at any age but are most common in older adults [1,2,3]. Patients under 70 years of age with GB showed a median life expectancy of around 14.6 months, even with the best current standard of treatment, which includes adjuvant chemotherapy with temozolomide (TMZ) and chemoradiotherapy after tumor excision. Population-based research indicates that chances of survival decline with age [4,5,6].

Considering most GB patients pass away from the illness in under a year and almost none survive long-term [7], GB tumors have attracted a lot of interest in the research community [8,9,10]. Even after extensive surgery, concomitant radiation, adjuvant TMZ, and rigorous treatment, the median survival period for adult patients is still only around 10 months, which may become up to 14 months with combination treatment and radiation; just 3–5% of patients live longer than three to five years after diagnosis [11,12,13,14,15]. In addition to patient characteristics like age and gender, various molecular markers such as mutations in IDH 1/2 (isocitrate dehydrogenase) [16], codeletion of chromosome arms 1p and 19q (1p19q co-del) [17], mutations in the telomerase reverse transcriptase promoter (TERT-p) [18], mutations in the MGMT promoter [19], and mutations in the EGFR gene [20] may impact disease susceptibility and progression [21], which makes predicting GB survival a challenging task.

IDH enzymes, consisting of three isoforms, are vital in several significant metabolic processes, including the Krebs cycle, glutamine metabolism, lipogenesis, and redox regulation [22]. More than 80% of World Health Organization (WHO) Grade 2/3 cases of glioma are characterized by IDH mutations [23]. In instances of WHO Grade 4 glioblastoma (GB), IDH mutations are often seen in secondary GB, which comprises 73% of clinical cases, and are less prevalent in primary GB (3.7%) [24].

Diffuse gliomas with mutations of the promoter region of TERT (TERT-p) have ambivalent prognoses. TERT-p mutations are frequently found in low-grade and glioblastoma tumors, but they are associated with a contrary prognosis. In low-grade gliomas (Grades 2 and 3), TERT-p mutant patients have a better prognosis than wildtype patients, but in GBs (Grade 4), TERT-p mutations are associated with poor outcomes [25,26]. Approximately 83% of primary GBs [27] have a specific mutation where a C nucleotide is changed to a T nucleotide in the promoter region of the TERT gene [28]. As a consequence of the mutation, GABPA (GA-binding protein A) recognizes an additional binding site for ETS (E26 transformation-specific family transcription factor), which facilitates the reactivation of telomerases [29].

Numerous studies have linked MGMT gene promoter methylation to chemotherapy response, especially alkylating medications like temozolomide [30,31,32]. The MGMT gene on chromosome 10q26 produces a DNA-repair protein that removes alkyl groups from guanine’s O6 position, a crucial DNA alkylation site. MGMT is implicated in DNA repair and glioma cell alkylating drug resistance [33,34]. Hypermethylation of CpG islands in the promoter region suppresses MGMT activity in glioma cells [35]. Hyper-methylation of the MGMT promoter improves OS and TMZ responsiveness in GBM patients [36,37].

Nearly 40–50% of GB patients had EGFR amplification detected by next-generation sequencing (NGS), while 14.4–26% have EGFR mutations [38,39]. The Consortium to Inform Molecular and Practical Approaches to CNS Tumor Taxonomy (cIMPACT-NOW) proposed a diagnostic entity for Grade 2 and 3 isocitrate dehydrogenase (IDH)-wildtype astrocytoma, which resembles GB due to TERTp mutation, EGFR amplification, or a combination of whole chromosome 7 gain and 10 loss (+7/−10) [40]. As of 2021, only IDH wildtype tumors are classified as GB in the 5th edition of the WHO classification of CNS tumors. The previously classified GBs with IDH mutations have been reclassified as astrocytomas with IDH mutations in grade 4 [41]. The molecular characteristics of this novel tumor type make EGFR a potential diagnostic and prognostic biomarker [42].

Age at diagnosis, surgical resection, tumor grade, chemo/radiotherapy, and genetic profiling affect glioma survival. This study evaluated the characteristics of glioma patients treated at Sina Hospital in Tehran over the past decade. We highlighted and discussed critical issues such as the importance of molecular profiling in predicting OS, the association of glioma survival with prognostically favorable clinical factors, and future research priorities for this patient population.

## 2. Materials and Methods

### 2.1. Population Characteristics and Study Design

This retrospective study involved patients at Sina Hospital in Tehran, Iran, from March 2012 to September 2020. The Ethics and Research Committee of Tehran University of Medical Science, Neurosurgical Department of Sina Hospital (IR.TUMS.SINAHOSPITAL.REC.1399.111) approved this project. Included patients had a confirmed diagnosis of glioma based on histological examination. In addition, pediatric patients were excluded. Various data points were collected, including age, gender, extent of tumor resection, tumor location, chemo/radiotherapy details, and genomic profiling information such as IDH1 status, MGMT-met, TERTp-mut, and EGFR-amp. Data were obtained by examining patient hospital records and using a data collection form. The classification of all tumors was performed according to the WHO classification system. The male-to-female ratio and average age at diagnosis were calculated for each histological subtype. To maintain ethical standards, patient information was extracted from archived records using pseudonyms to ensure the confidentiality of participants. This study does not include any patient-specific information. Patients who underwent emergency operations and those with a Karnofsky Performance Status (KPS) score below 70 and/or a history of psychiatric diseases were excluded (please see Figure 1 for further details).

### 2.2. Genetic Profile

Formalin-fixed GB tumor samples were encompassed with paraffin selected for DNA extracting according to the Reinfenberger et al. study [43]. Genetic profile information included (1) MGMTp-met, (2) IDH1-mut, (3) TERTp-mut, and (4) EGFR-amp registered in the database.

(1) MGMT promoter methylation (MGMTp-met) was analyzed and recorded by methylation-specific polymerase chain reaction (PCR) according to the reported data by Mollemann et al. [44]. The primer sequences used to detect methylated MGMT promoter sequences were “5-GTT TTT AGA ACG TTT TGC GTT TCG AC-3 and 5-CAC CGT CCC GAA AAA AAA CTC CG-3”. The primer sequences used to detect unmethylated MGMT promoter sequences were “5-TGT GTT TTT AGA ATG TTT TGT GTT TTG AT-3 and 5-CTA CCA CCA TCC CAA AAA AAA ACT CCA-3”. (2) IDH1 specific part of exon 4, comprising the R132 mutation hotspot, was amplified from genomic DNA by polymerase chain reaction (PCR), and the high-resolution melting curve analysis (HRM) was followed by sequence analysis [45]. Previous worthwhile studies reported amplifications of a 122 bp base pairs length fragment spanning IDH1 [46]. Based on the HRM guidance on a Light Cycler 480, HRM analysis was performed, and the result was entered into our study database. (3) Real-time quantitative PCR (qPCR) using the Light Cycler 480 format recognized the mRNA expression levels of TERT, which was reported before by Arita et al. [47]. Moreover, Light Cycler 480 was used in relative quantification analyses. TERT-specific primers, which are located in exon 5, were used from formalin-fixed paraffin-embedded samples: “GCCTGAGCTGTACTTTGTC” (P0155), and the reverse primer on exon 6: “CGTGTTCTGGGGTTTGATG” (P0156). TERT mRNA expression measurement was incompatible with human total brain RNA.

(4) According to the manufacturer’s recommendations, 12 μL of the RNA isolated from 1 mL of CSF was reverse-transcribed using the Superscript VILO cDNA synthesis kit (Invitrogen (Waltham, MA, USA)). Samples were then preamplified using the TaqMan PreAmp Master Mix (Applied Biosystems (Norwalk, CT, USA)). Briefly, 12.5 μL of the cDNA was added to the PreAmp Master Mix together with all the genes of interest and preamplified for 14 cycles according to the manufacturer’s recommendations. The samples were then diluted 1:10, and TaqMan quantitative reverse transcription PCR was performed on all samples for all the selected genes. The amplification was performed using ABI PRISM 7500 with the following program: 50 °C, 2 min; 95 °C, 10 min; 40 cycles of 95 °C, 15 s, 60 °C, 1 min on standard mode. Logarithmic amplifications were interpreted as positive, and relative quantities versus GAPDH/18S were reported for each analyzed sample. Wildtype EGFR primer sequences: “5′-TATGTCCTCATTGCCCTCAACA”. “3′-CTGATGATCTGCAGGTTTTCCA”. EGFRvIII primer sequences: “5′-CTGCTGGCTGCGCTCTG”. “3′-GTGATCTGTCACCACATAATTACCTTTC”. To prepare the templates for Sanger sequencing, genomic DNA was amplified using the BigDye Terminator Cycle Sequencing Kit v3.1 with the same primer pair as pyrosequencing without biotinylating the reverse primer [48].

### 2.3. Statistical Analysis

The primary focus of our study was to assess the impact of specific clinical variables and molecular profiles on overall survival (OS), defined as the duration from the day of tumor diagnosis to the date of death from any cause. To gain insights into the characteristics of the patients, we conducted a thorough descriptive analysis. Descriptive statistics were employed to summarize continuous measures, including the number of observed values, median, standard deviation, median, and range. To examine the differences between subgroups, categorized data were compared using the Chi-square (*χ*^2^) test. Furthermore, we employed the two-tailed Student’s t-test to analyze the age distribution comparison. The reverse Kaplan–Meier method was utilized to estimate the median survival time. Survival curves were generated using Kaplan–Meier estimates, and differences between the curves were analyzed using the log-rank test. A two-tailed *p*-value < 0.05 was considered statistically significant. We used Cox proportional hazards regression analysis to evaluate the impact of measured variables on patient survival in multivariate adjusted models, using a stepwise Wald backward selection procedure. All statistical analyses were performed using *R* software version 4.1.0.

## 3. Results

A complete flow diagram of patient selection is provided in Figure 1. This study investigated the association between OS and various factors, including gender, age, distinct tumor grades, the extent of resection, multiple surgeries, radio- and chemotherapy received, mutational profiles in IDH1 and TERTp, EGFR amplification, and MGMTp-met. Among the cohort of 178 patients meeting the inclusion criteria, a combined treatment approach incorporating radiation therapy and chemotherapy was administered to 56 patients. Additionally, 11 patients received radiation therapy as a standalone treatment, while 5 patients exclusively underwent chemotherapy.

### 3.1. Association between Glioma Grades and Parameters Studied

Figure 2 provides an overview of the prevalence of genetic and epigenetic alterations identified in the 178 glioma patients with available follow-up. A detailed description of the clinical characteristics and molecular profiling observed in the patients can be found in Table 1. The median ages of patients with Grade 2, 3, and 4 tumors were 40.8, 45.9, and 46.5 years, respectively. This patient cohort with newly diagnosed glioma comprised 115 (64.6%) males and 63 (35.4%) females, as shown in Figure 3. The *χ*^2^ value was estimated to compare the expected and observed ratios for sex groups, age groups, tumor grade, radiotherapy, and chemotherapy across different tumor grades. Both men and women demonstrated an increase in mortality with an increasing glioma grade (Grade 4) (Figure 4A). An analysis of patient age and mortality rates revealed that in both age groups (age < 50 and age ≥ 50), the mortality rate was significantly higher in patients with Grade 4 glioma compared to the other groups (Figure 4B). The mortality rate of patients with Grade 4 tumors in both the unilobar and multilobar groups was significantly higher than in patients with Grade 2 and 3 glioma (Figure 4C). Only patients with Grade 4 tumors who underwent chemotherapy showed significantly lower mortality rates and higher survival than patients who did not (Figure 4D). Similarly, patients with Grade 4 tumors subjected to radiotherapy showed a higher survival rate compared to those not treated (Figure 4E); both chemo- and radiotherapy appeared to improve survival in Grade 3 patients as well, although these effects did not reach statistical significance (Figure 4D,E).

### 3.2. IDH1 and EGFR Status

In this study, the IDH1 and EGFR status was evaluated in tissue samples from each patient (Figure 2), and the IDH1 and EGFR-amp were determined (Table 2). It should be noted that genetic profile information was available for 127 out of the total 178 patients included in the study. Among the Grade 2 gliomas (*n* = 26), 17 cases (65.3%) exhibited IDH1 mutations, while 3 patients (11.5%) had EGFR-amp. In grade 3 gliomas (*n* = 26), 22 cases (84.6%) showed IDH1 mutations, and no EGFR-amp was observed. In Grade 4 gliomas (*n* = 75), 61 patients (81.3%) had IDH1 mutations and 1 patient (1.3%) had an EGFR-amp. Among the 26 grade 3 glioma samples, 4 cases (15.3%) had wildtype IDH1, and 26 cases (100%) had no EGFR-amp. On the other hand, in the 75 Grade 4 glioma cases, 14 cases (18.6%) had wildtype IDH1 and 74 cases (98.6%) had no EGFR-amp. It is worth noting that the frequency of IDH1 mutations was higher compared to EGFR-amp in Grade 2, 3, and 4 gliomas. Specifically, we observed only 3, 0, and 1 EGFR-amp among Grade 2, 3, and 4 gliomas, respectively.

### 3.3. TERT and MGMT Genes and Clinical Response in Patients

In the study cohort of 127 cases, the IDH1, MGMT, TERT, and EGFR statuses were evaluated successfully. Among these cases, 100 (78.7%) exhibited IDH1 mutations, 20 (15.7%) had TERTp-mut, 89 (70.0%) were classified as MGMTp-met tumors, and 4 (3.1%) showed EGFR-amp. MGMTp-met rates were lower (6 patients) in patients with IDH1 wildtype tumors and higher (83 patients) in patients with IDH1 mutant tumors. Of the 178 patients with treatment recorded, 61 (34.3%) and 67 (37.6%) received TMZ and radiotherapy, respectively. As of the time of final data collection, 56 patients (31.5%) were still alive or lost to follow-up.

In this study of 166 patients, tumors in the frontal lobe were the most prevalent (93 patients), followed by tumors in the temporal lobe (73 patients), parietal lobe (68 patients), occipital lobe (27 patients), and insular cortex or cerebellum (17 patients). The most common tumor combinations were observed in the frontal + temporal (18 patients), parietal + frontal (15 patients), and parietal + frontal + temporal (15 patients) lobes (Figure 5A). None of the patients had tumors in all four brain lobes simultaneously. Occipital lobe tumors tended to manifest as isolated cases in glioma patients, with 10 out of 27 cases showing no signs of tumors in other brain lobes (Figure 3 and Figure 5A). Anatomically, the location of tumors was primarily multilobar (83 of 166 patients), with a subset of patients demonstrating frontal (21.6%, *n* = 36) and temporal (9.0%, *n* = 15) gliomas. Other tumor locations included the parietal lobe (10.2%, *n* = 17), insular cortex or cerebellum (3.6%, *n* = 6), and occipital lobe (6.0%, *n* = 10) (Figure 5A,B). The tumor location was unknown in 12 patients due to unavailable clinical follow-up and imaging data.

The distribution of MGMT methylation and unmethylation frequencies demonstrated distinctive patterns across different tumor locations within the brain (Figure 5C,D). Our analysis involved 127 glioma patients, revealing a significant prevalence of MGMTp-met in the multilobar tumors (30.7%, 39/127), followed by the frontal lobe (15.0%, 19/127) and parietal lobe (8.7%, 11/127). Importantly, no instances of MGMT unmethylation were identified in the insular or cerebellum lobes and the temporal lobe. We conducted a chi-square test to explore the association between MGMT status and tumor location, which yielded a statistically significant correlation (*χ*^2^ = 13.77, *p*-value = 0.048) between these variables (Figure 5C). Furthermore, the distribution of TERTp-mut and TERTp-wt frequencies varied depending on the tumor location within the lobe. In CNS tumors, TERTp-wt (70.1%, 89/127) was more prevalent than TERTp-mut (29.9%, 38/127). Among specific lobes, the multilobar region displayed the highest frequency of TERTp-wt (35.4%, 45/127), followed by the frontal lobe (21.3%, 27/127), and parietal lobe (8.7%, 11/127). No mutations were detected in the insular or cerebellum lobes and the occipital lobe. Figure 5D shows the association between TERT-p status and tumor location based on the lobe in CNS tumor patients. Notably, the chi-square test revealed a significant correlation (*χ*^2^ = 14.62, *p*-value = 0.042) between these variables. A meticulous examination of the data highlights the impact of the TERT-promoter status and MGMT methylation on the specific lobes, offering valuable insights into the molecular characteristics of CNS tumors. A comprehensive analysis of the data reveals significant implications of the TERT-p status and MGMT methylation on distinct lobes, providing valuable insights into the molecular characteristics of CNS tumors.

The alluvial plot (Figure 6) shows the distribution and relationships between survival rate, tumor grade, TERT genetic profile, and adjuvant radiotherapy. A considerable number of patients with tumor grade 4 (*n* = 86) had a high mortality rate (88.6%). Furthermore, the chart illustrates that patients who received adjuvant radiotherapy had a significantly lower mortality rate. The association between the TERT genetic profile and survival rate further underscores the relevance of genetic characteristics in prognostic evaluations.

### 3.4. Survival Analysis

Results of multivariate Cox regression analyses for clinical characteristics and molecular profiles, including IDH1, TERT, and MGMT, are presented in Table 3. In multivariate models, the histological grade of the tumor (HR, 5.82; 95% CI, 3.53–9.58; *p* < 0.0001), adjuvant radiotherapy (HR, 1.87; 95% CI, 0.37–2.70; *p* < 0.018), IDH status (HR for IDH1 mutant status, 2.25; 95% CI, 0.99–3.86; *p* < 0.043), TERT status (HR for unmethylated status, 0.889; 95% CI, 0.290–1.19; *p* < 0.0143) were significantly associated with OS (Table 3).

### 3.5. Factors Associated with Overall Survival

The median survival of patients was 20 months (600 days; Figure 7A), with 15.4% and 61.0% of patients alive at 12 and 24 months, respectively. In total, 19 patients (15.3%) survived for longer than 48 months (4 years) and were considered long-term survivors. Interestingly, the four patients who were still alive when the dataset was finalized (OS > 50, 53, 65, and 71 months) exhibited IDH1-mut and MGMTp-met. Initially, we assessed whether any genetic alterations were associated with survival through univariate and multivariate analyses (Table 3). A more stringent multivariate analysis, which only incorporates parameters with *p* > 0.05 in the univariate analyses, revealed the same parameters were independent prognostic factors. The distributions of the survival curves are shown in Figure 7B–I. The survival analysis using Kaplan-Meier estimation did not demonstrate any significant differences in survival among glioma patients based on sex, extent of resection, and tumor location (Figure 7D–F). Glioma patients with IDH1-mut had a median OS of 21 months compared to 17.5 months for those without an IDH1 mutation (Figure 7G). Methylation of MGMT was associated with a median OS of 21 months compared to 19 months in non-methylated cases (Figure 7I). Patients with a TERTp-mut exhibited a median OS of 16 months, whereas patients with a non-mutated TERT promoter had a median OS of 21 months (Figure 7H). Notably, patients with solely an IDH1 mutation exhibited the most favorable survival, with a median survival of 21 months. Subsequently, patients with IDH1-mut and MGMTp-met experienced a slightly lower median survival of 20.5 months. In contrast, patients lacking either an IDH1-mut or MGMTp-met had the shortest median survival of 16 months. It is worth noting that the molecular profile analysis, specifically regarding wildtype IDH1 and MGMT methylation, encompassed a limited sample size of only three patients who exhibited survival durations of 18, 50, and 62 months, respectively. Furthermore, we assessed the prognostic significance of histological grade and clinical characteristics. Among the histological subgroups, Grades 2 and 3 demonstrated a more favorable prognosis, with median OS durations of 52.5 and 47 months, respectively. Conversely, Grade 4 was associated with an extremely poor outcome, with an OS of 18 months (Figure 7B). Additionally, patients under the age of 50 displayed a more favorable prognosis, with a median OS of 24 months, compared to the subgroup aged 50 years and above, which exhibited a median OS of 19 months (Figure 7C).

## 4. Discussion

Our results showed a difference of 42, 45, and 46 years in the median age of patients with Grade 2, 3, and 4 tumors, respectively. Gliomas were more common in males than in females, and the death rate increased with glioma grade, with the highest mortality in grade 4 patients. GB is the most prevalent malignant brain tumor in adults, accounting for 54% of all cases. The incidence of GB increases with age, and the development of illnesses, especially those affecting the CNS, is more likely among older people. Wildtype IDH GB is the most common aggressive primary brain tumor in adults, with an average diagnostic age of 68–70 years, and progresses rapidly. Elderly (65 and older) GB patients may have a worse post-treatment survival rate due to age-related changes, such as diminished immune system function and persistent neuroinflammation [35,49]. A negative correlation exists between the age at which GB is identified and the prognosis. The 10-year relative survival rate declines from 15.2% to 5.7% between 20–44 and 45–54 years of age, respectively [50,51].

The results of the current study show that glioma patients with IDH1-mut had a median OS of 21 months versus 17.5 months for those without an IDH1 mutation. In addition, MGMTp-met was associated with a median OS of 21 months versus 19 months in patients with a non-methylated. The combination of IDH1-mut and MGMTp-met status is a more accurate predictor of survival in glioblastoma compared to either IDH1 or MGMT alone. Glioblastoma patients were categorized into 3 distinct genotypes based on the genetic and epigenetic characteristics of IDH1 and MGMT: GB patients with mutant IDH1/MGMT-met had the longest survival, followed by patients with mutant IDH1/MGMT-unmet or wildtype IDH1/MGMT-met, and patients with wildtype IDH1/MGMT-unmet had the shortest survival [52]. Improved prognosis has been linked to tumor molecular characteristics such as MGMTp-met and IDH1 mutation [34,52,53]. Thus, MGMT methylation positively impacts survival and responsiveness to TMZ therapy [53]. In addition, IDH and MGMT co-methylation are linked to a better prognosis and predict the response to chemotherapy and surgical resection [52].

Most Grade 4 gliomas showed IDH1 mutations in the present study. Unlike most Grade 4 astrocytomas, these have a greatly different epidemiology. There is still uncertainty regarding the relative importance of genetic and environmental risk factors in gliomagenesis. According to research, however, cancer-causing mutations in gliomas come primarily from endogenous sources rather than exogenous ones [54]. There is a possible association between primary CNS tumors and ionizing radiation, some toxic agents (N-nitroso compounds, pesticides), air pollution, and radiofrequency electromagnetic waves. A well-established risk factor for brain tumors is brain ionizing irradiation, especially in childhood. There has been scanty exploration of exposure to environmental toxins and even prenatal exposure to N-nitroso compounds results in brain tumors. Large prospective studies contradict outdoor pollution and brain tumor risk. In adults, the effects of mobile phones on brain tumor risk have not been established for glioma and meningioma [55].

Surgical resection, radiation, and TMZ chemotherapy constitute the standard treatment for GB. Despite the intensive nature of this therapy, the tumor reoccurs within 7 to 10 months following surgery in 75–90% of GB patients [56,57,58]. A study utilizing molecular testing revealed distinct genetic profiles among GB survivors. Notably, mutations in the IDH genes (IDH1 and IDH2) and methylation of the MGMT promoter were identified as two significant factors associated with a more favorable response to standard clinical care [59]. Patients with mutant IDH1/2 GB exhibited superior outcomes to those with wildtype IDH tumors, with 42- and 14-months survival, respectively [60]. The literature suggests a strong correlation between TERT gene polymorphisms and an increased risk of glioma in patients [61]. TERT mutations are also linked to biomarkers such as IDH1, 1p19q, TP53, and EGFR [62].

Gliomas are more prevalent in men than in women. The tumors of glioblastoma patients showed huge genetic sex differences linked to survival. According to a study on differentially expressed genes in the tumor clusters, survival in men was impacted the most by genes that govern cell division. Regarding survival in females, integrin gene expression was the most critical mechanism for tumor dissemination [61]. Other work indicated that male patients had the lowest cancer-specific survival (CSS) rates throughout localized cancer stages and various age groupings, which was confirmed by stratified analysis [63,64].

One study reported that women with IDH1-mutant tumors were concentrated in the group with the most favorable prognosis, while in males, the mutations were spread out across all groups. Considering IDH1 mutations have been linked to higher survival in glioblastoma patients, the longest-surviving female cluster is consistent with this theory. However, this was not the case for men [63]. Sex variations in disease incidence and prognosis are well acknowledged but seldom understood enough to permit sex-specific therapy. Endocrinology and cancer research shows gonadal steroid hormones contribute to GB development and prevalence [65]. According to some studies, females have a longer survival rate than males [66,67]. In an orthotopic model of glioblastoma, Barone et al. [68] demonstrated that estrogen increased survival, and a study based on estradiol may be beneficial for treating GBM. Observations by Li et al. [69] indicate that estrogen protects patients from GBM by methylating estrogen receptors. Yu et al. [70] also found that androgen receptor signaling promoted GBM tumorigenesis by inhibiting TGF-β (transforming growth factor β) receptor signaling. Another study suggests estrogen may protect against GBM genesis and promote a more favorable biology once GBM occurs [64].

Our present study indicates that patients with a mutation in TERT had a median OS of 16 months compared to 21 months in those without a TERTp-mut. In addition, MGMTp-met status was shown to affect the prognostic value of a TERTp-mut. Only TERTp-mut GB with MGMTp-met may respond to TMZ; therefore, TMZ may not benefit all patients with MGMTp-met GB [71]. Long-term follow-up of patients with TERTp-mut GBs showed a high correlation between the prognosis and the presence of multifocal/distant lesions. There was an association between EGFR amp/gain, CDKN2A deletion, and PTEN loss as well as a negative correlation between CDK4 and TP53 deletion for the TERTp mutation [72]. EGFR mutations have been demonstrated to be effective prognostic indicators of OS in IDH-wildtype GBM patients. Moreover, data suggests that EGFR amplification is evident in high-grade gliomas (25%). EGFR amplification was also observed to be limited to IDH wildtype (26%) and TERT mutant (27%) gliomas, occurring irrespective of MGMT promoter methylation status and being mutually exclusive with 1p/19q co-deletion (LOH) [73]. The functional connection between EGFR and p53 in GBM is intriguing. Thus, EGFR has been shown to reduce the activity of wildtype p53 by increasing the interaction between DNA-PKcs and p53. Either EGFR or DNA-PKc knockdown enhanced the transcriptional activity of wildtype p53 due to a reduced interaction between p53 and DNA-PKcs. These results revealed a unique non-canonical regulatory axis between EGFR and wildtype p53 in GBM, with unexpected biological functions [74].

We observed an apparent difference (although not statistically significant) in the OS of gross (GTR) versus near-total resection (NTR) glioma patients, which is consistent with findings by Abdelfath et al. [75]. GTR appears to be more beneficial than subtotal resection (STR) in extending the life of elderly individuals with high-grade glioma [76]. Aggressive surgical resection should be considered for older GBM patients, especially those with relatively low KPS. Intraoperative magnetic resonance imaging (ioMRI) does not seem to provide any significant advantage over intraoperative ultrasonography (ioUS) in experienced hands in this population. Still, it may significantly prolong the duration of surgery, which is a modifiable prognostic factor that affects care [77]. Other work showed a correlation between maximal tumor excision and OS in all categories of glioblastoma patients. Additionally, maximal resection of non–contrast-enhanced (NCE) tumors was associated with longer OS in younger patients independent of IDH status and in patients with IDH–wildtype glioblastoma regardless of the methylation status of the promoter region of the DNA repair enzyme O6-methylguanine-DNA methyltransferase [78].

Within our cohort, the anatomical tumor location was generally confined to a multi-lobe, primarily multilobar disease (50.0%), with some patients demonstrating tumor growth in the frontal (21.6%) and temporal (8.4%) lobes. Other tumor locations included the parietal lobe (10.2%), occipital lobe (6.0%), and insular cortex or cerebellum (3.6%). Most glioblastomas develop in the periventricular white matter areas close to the subventricular zone. MGMTp-met tumors are more prevalent in the left temporal lobe, particularly in patients with GB, an IDH1 mutant tumor, tumors with the proneural gene expression subtype, or frontal lobe tumors missing PTEN deletion. IDH1 mutation-associated MGMTp-met tumors tended to manifest in the left frontal lobe, whereas EGFR-amplified and EGFR variant 3-expressing tumors occurred most frequently in the left temporal lobe. A comparable area in the left temporal lobe was associated with excellent radiochemotherapy response and improved survival [79]. In another study, tumors in the right occipitotemporal periventricular white matter were substantially related to poor survival in both training and test cohorts and had a greater tumor volume than tumors in other regions. Right parietal tumors were associated with hypoxia pathway enrichment and platelet-derived growth factor receptor (PDGFRA) amplification, deeming these processes appropriate subgroup-specific treatment targets. Additionally, central tumor placement was associated with a worse prognosis. In elderly individuals, the distance from the center of the third ventricle to the contrast-enhancing tumor border may be a practical prognostic indicator [79,80,81].

The location of the glioblastoma may be used as a predictor of the status of the TERT promoter mutation. Studies have shown that TERTp-mut gliomas are more likely to occur in the frontal or temporal lobes [82,83]. Most IDH-mutated gliomas were located in a single lobe, such as the frontal lobe, temporal lobe, or cerebellum, and rarely in the diencephalon or brain stem. Additionally, IDH-mutated tumors were rarely found in high-risk brain regions where surgery has a high intraoperative and postoperative mortality rate [84].

Our study recognizes certain limitations, including using data from a single center and an underpowered analysis for detecting differences among patient subgroups. The scarcity of data on comorbidities such as lung infections, renal disorders, seizures, high blood pressure, and paresthesia emphasizes the necessity for more extensive and standardized data collection. Moreover, we did not adjust for multiple comparisons. In the current study, another limitation was the lack of additional tests, such as CDKN2A/B and 1p19q status, which could lead to poorer outcomes. However, the study’s strength lies in its meticulous examination of factors associated with glioma prognosis and treatment response, presenting valuable insights for clinical decision-making. Additionally, we highlighted the synergistic impact of specific genetic profiles and radiotherapy, identifying potential targets for personalized treatment strategies. Despite the acknowledged limitations, this study comprehensively analyzes crucial factors influencing glioma prognosis and treatment response.

## 5. Conclusions and Future Direction

A disappointing fact is the long-term lack of development in glioma prognosis and therapy. Predicting clinical outcomes is challenging due to subjective criteria and the difficulty of distinguishing the heterogeneous histological appearance of glioma tissues. Conventional histological diagnosis may be hampered by morphological ambiguity and interobserver disagreement. It is crucial to create a reliable and objective molecular marker for identification. The present study aimed to investigate the impact of clinical and molecular factors on the overall survival of glioblastoma patients. Multivariate Cox regression analysis revealed that histological tumor grade, adjuvant radiotherapy, IDH status, and TERT-p status were significantly associated with overall survival. Notably, patients with IDH1 mutations and TERTp-wt exhibited more prolonged survival. Factors such as extensive tumor removal, smaller tumor size, and prompt initiation of radiation therapy after surgery were associated with favorable prognoses. Patients with higher tumor grades had poorer outcomes, while those who received adjuvant radiotherapy showed improved survival. In conclusion, this study highlights the importance of molecular and clinical characteristics in predicting overall survival in glioblastoma patients and provides valuable insights for personalized treatment strategies. As understanding of the molecular underpinnings of glioblastoma continues to develop, molecular markers will become increasingly important in prognosis and clinical decision-making. This devastating disease can only be managed and treated more effectively by integrating molecular profiling into routine clinical practice and conducting ongoing research.

## Figures and Tables

**Figure 1 cancers-16-02121-f001:**
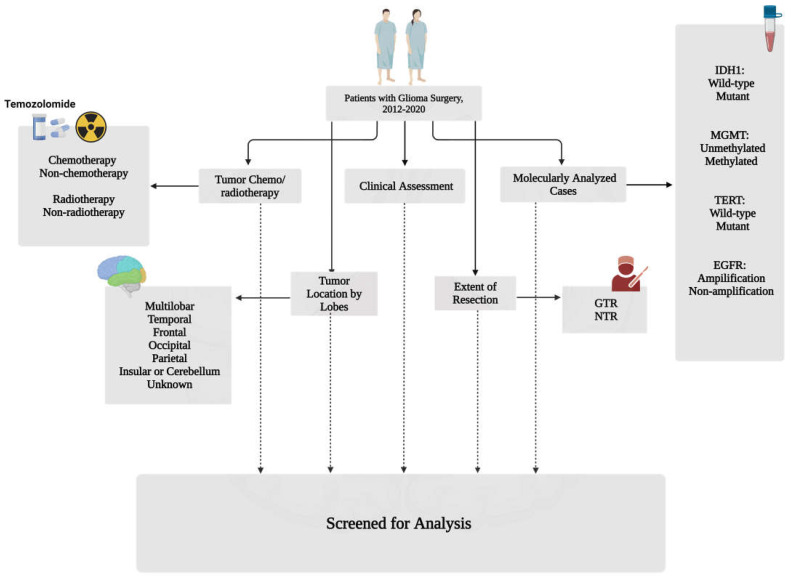
Study design and flow chart of patient selection (created with BioRender.com).

**Figure 2 cancers-16-02121-f002:**
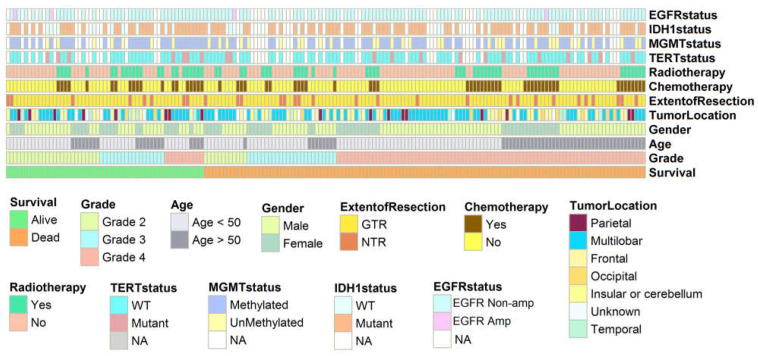
The distribution of molecular and clinical characteristics in the 178 glioma patients. Molecular classification was arranged by the overall survival of patients (alive and dead).

**Figure 3 cancers-16-02121-f003:**
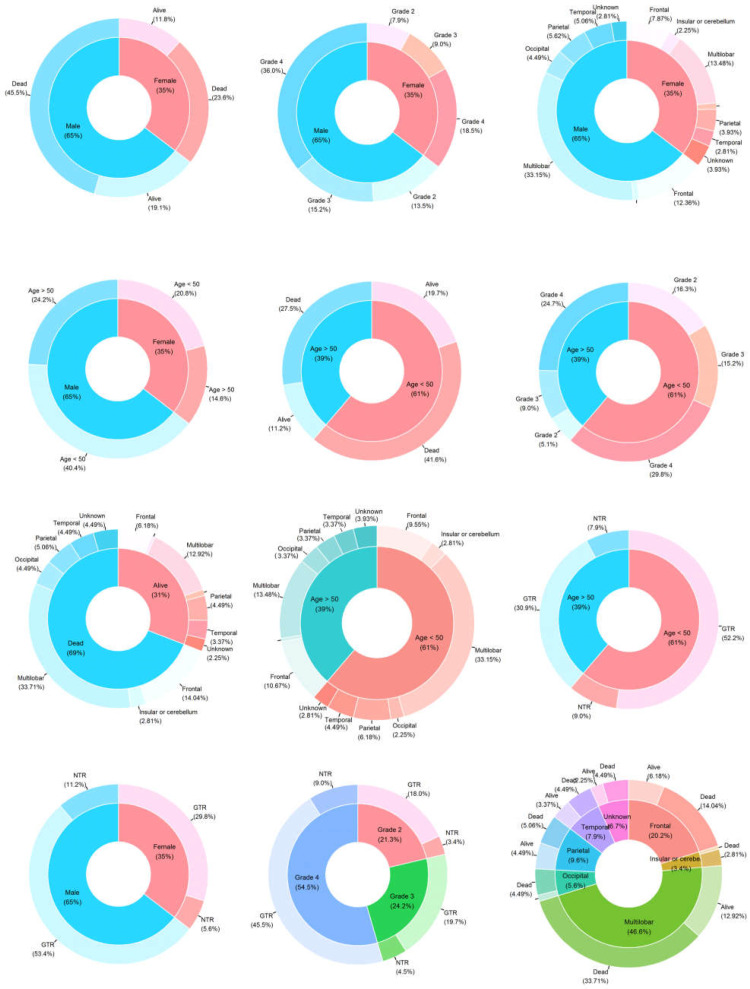
Pie charts showing the frequency of specific characteristics among the 178 patients of the studied glioma cohort. The area of each segment is proportional to the number of patients.

**Figure 4 cancers-16-02121-f004:**
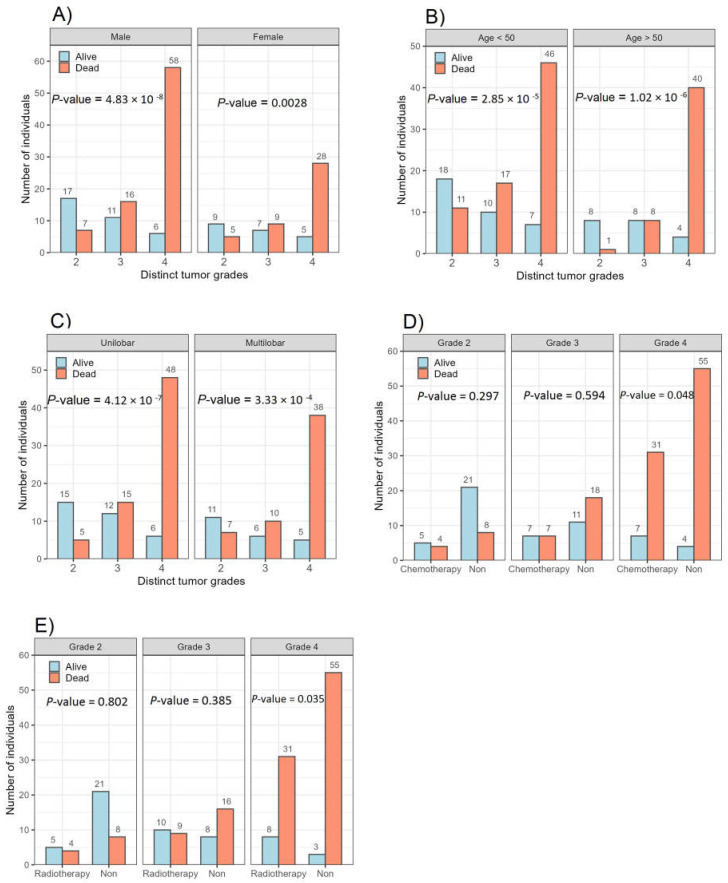
The ratios of different grades of glioma in the examined groups are shown in (**A**) gender vs. survival, (**B**) age groups vs. survival, (**C**) tumor location vs. survival, (**D**) chemotherapy vs. survival, and (**E**) adjuvant radiotherapy vs. survival. The chi-square test was used to compare the groups.

**Figure 5 cancers-16-02121-f005:**
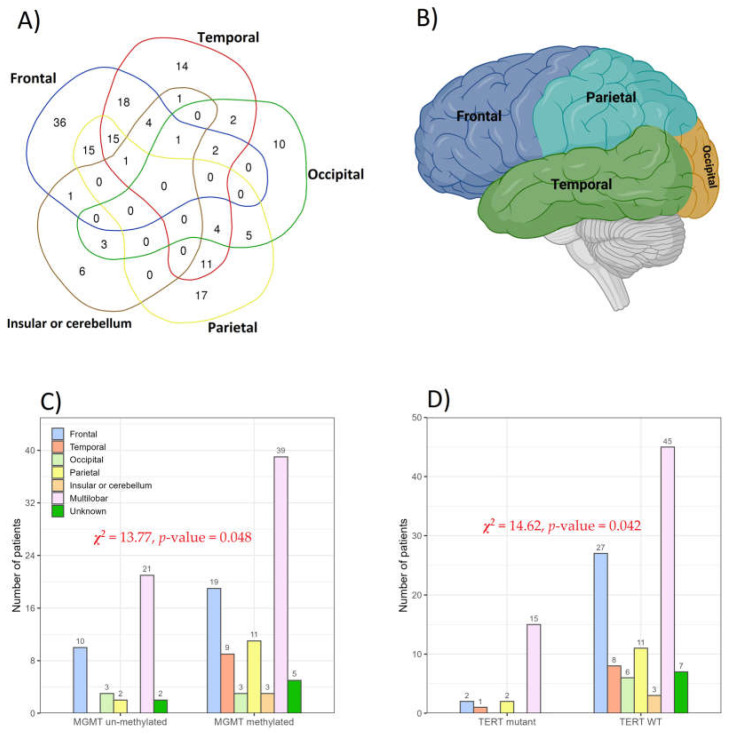
(**A**) Venn diagram illustrating the distribution of 167 patients with tumors in the “Temporal”, “Frontal”, “Occipital”, “Parietal”, and “Insular or cerebellum” lobes of the brain. (**B**) The lobes of the brain (created with BioRender.com). The association between (**C**) MGMT promoter methylation and (**D**) TERT-promoter status with tumor location is based on the lobe investigated in this study.

**Figure 6 cancers-16-02121-f006:**
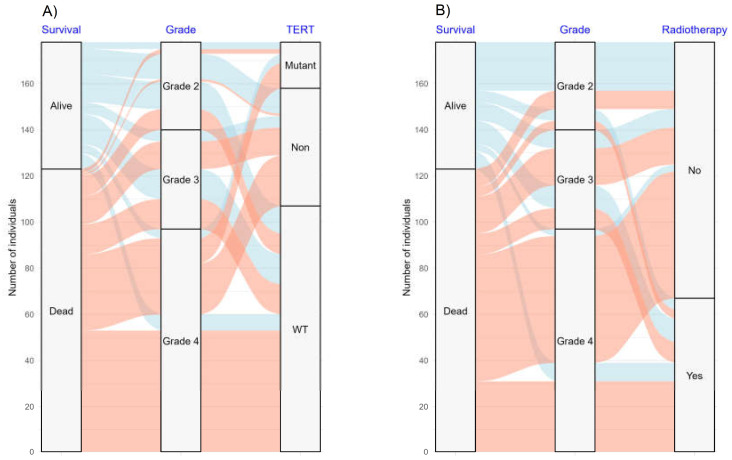
The alluvial plot shows the distribution of patients, survival subgroup, and histological grade in relation to (**A**) TERT and (**B**) adjuvant radiotherapy.

**Figure 7 cancers-16-02121-f007:**
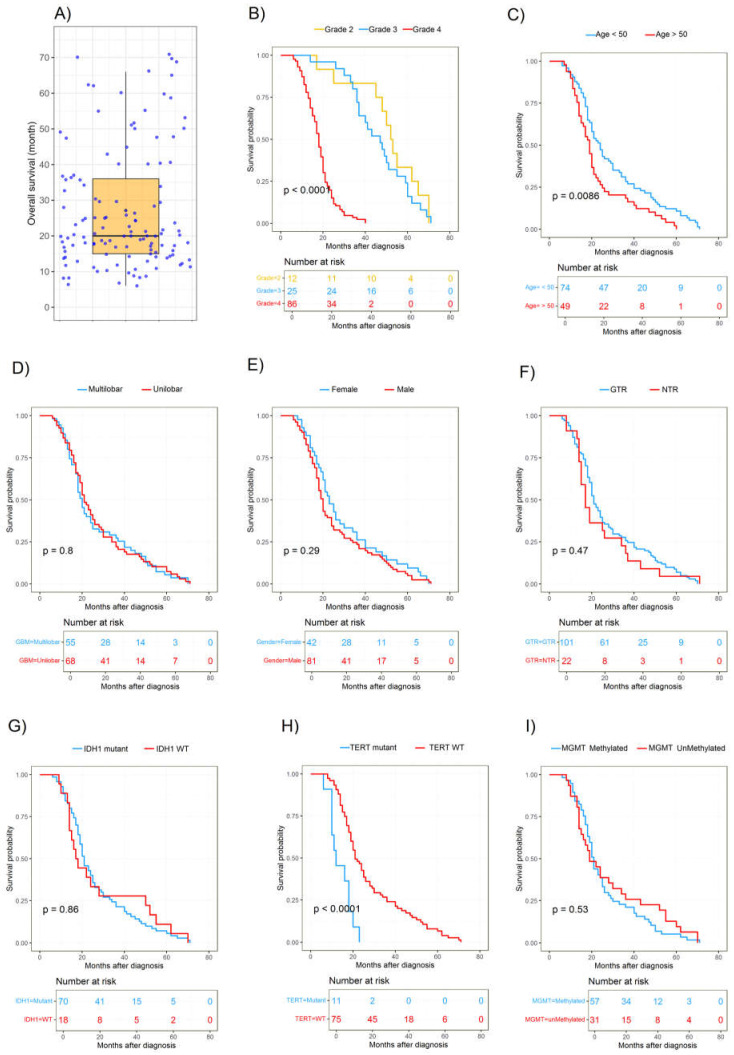
The distribution of glioma patient survival and Kaplan–Meier curves for overall survival are shown for 8 risk groups. (**A**) The box plot illustrates the overall survival (months) of glioma patients. Kaplan–Meier curves show survival probabilities in relation to (**B**) the histological grade, (**C**) age groups, (**D**) uni- and multilobar glioma, (**E**) gender, (**F**) extent of resection, (**G**) IDH1 status, (**H**) TERT status, and (**I**) MGMT status.

**Table 1 cancers-16-02121-t001:** Characteristics of the 178 patients with glioma in this study.

Characteristics	Cases	%
Age group		
<50 years	109	61.2
≥50 years	69	38.8
	Median age (range); years	45 (22–81)
Gender		
Male	115	64.6
Female	63	35.4
Distinct tumor grades		
Grade 2	38	21.3
Grade 3	43	24.2
Grade 4	97	54.5
Extent of resection (EOR)		
Gross-total resection (GTR)	148	83.1
Near-total resection (NTR)	30	16.9
Tumor location by lobe		
Multilobar	83	46.6
Temporal	14	7.9
Frontal	36	20.2
Occipital	10	5.6
Parietal	17	9.1
Insular or cerebellum	6	3.4
Unknown	12	6.7
Survival		
Alive	55	30.9
Dead	123	69.1
Survival ≤ 12	19	15.4
12 < Survival ≤ 24	56	45.5
24 < Survival ≤ 36	18	14.6
Survival ≥ 36	30	24.4
Multiple surgeries		
Yes	60	33.7
No	118	66.3
Anesthesia		
Awake	20	12.1
General	146	87.9
Chemotherapy		
Yes	61	34.3
No	117	65.7
Adjuvant radiotherapy		
Yes	67	37.6
No	111	62.4
Radio/chemotherapy		
Yes	56	31.5
No	122	68.5
Type 2 diabetes mellitus		
Yes	14	7.9
No	164	92.1

**Table 2 cancers-16-02121-t002:** The distribution of IDH1, MGMT, TERT, and EGFR mutations among different histological grades of glioma in 127 patients with available genetic profiling.

Molecular Assay	Different Grades of Glioma Tumor			Total
	Grade 2 (26)	Grade 3 (26)	Grade 4 (75)	
IDH1 status				
Wild type	9	4	14	27
Mutant	17	22	61	100
MGMT methylation status				
Methylated	19	17	53	89
Unmethylated	7	9	22	38
TERT promoter status				
Wild type	21	26	60	107
Mutant	5	0	15	20
EGFR amplification status				
Non-amplified	23	26	74	123
Amplified	3	0	1	4

**Table 3 cancers-16-02121-t003:** Prognostic univariate and multivariate Cox regression analyses in glioma patients.

Characteristic	Patients Number	Univariate	Multivariate			
		*p*-Value	*p*-Value	HR	95% CI for HR	
					Lower	Upper
Age, per year (<50 vs. >50)	123	0.010 *	0.224	1.370	0.866	1.845
Gender (Male vs. Female)	123	0.288	0.530	1.148	0.773	1.707
Histological grade (Grades 2, 3, and 4)	123	2.2 × 10^−16^ *	8.57 × 10^−11^ *	5.823	3.539	9.580
Extent of resection (GTR vs. NTR)	123	0.513	0.331	1.291	0.787	2.118
Glioma (unilobar vs. multilobar)	123	0.764	0.311	0.806	0.531	1.224
Chemotherapy (Yes vs. No)	123	0.791	0.539	1.297	0.569	2.957
Adjuvant radiotherapy (Yes vs. No)	123	0.0057 *	0.0181 *	1.807	0.375	2.705
IDH1 status (WT vs. Mutant)	88	0.430	0.043 *	2.255	0.990	3.863
TERT mutation (WT vs. Mutant)	88	0.00041 *	0.0143 *	0.889	0.290	1.196
MGMT methylation (WT vs. Mutant)	88	0.769	0.553	0.848	0.493	1.461

* *p*-value ≤ 0.05.

## Data Availability

The authors confirm that the data supporting the findings of this study are available within the article.

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
