# Peer review of "Genetic Prognostic Factors in Adult Diffuse Gliomas: A 10-Year Experience at a Single Institution"

_cancers, 2024, doi:10.3390/cancers16112121_

Round 1
Reviewer 1 Report
Comments and Suggestions for Authors
The authors have done a retrospective study (single centre) on the glioma patient and evaluated IDH1, MGMT, and TERT mutation and also investigated EGFR status. They evaluated the correlation between OS in GBM patients and sociodemographic as well as clinical factors, and molecular profiling based on these results. It is an important study on the glioma including GBM and it is the first study in that area. So the results are important.
1- In the introduction, it is expected the authors explain about the impact of IDH1, MGMT, and TERT mutation in glioma . Right now the explanation is subminimal and need further expansion.
2- The hypothesis and objectives should be clearly stated in the introduction toward the end.
3- Figure 2 needs more explanation in the results. It is hard to follow up the results in the figure 2.
4- Is there any correlation between, MGMT, IDH1 and TERT mutation and with tumors in the “Temporal”,” Frontal”,” Occipital”,” Parietal” and “Insular or cerebellum” lobes of the brain
5-What could be the future application of this study?
Author Response
The authors have done a retrospective study (single centre) on the glioma patient and evaluated IDH1, MGMT, and TERT mutation and also investigated EGFR status. They evaluated the correlation between OS in GBM patients and sociodemographic as well as clinical factors, and molecular profiling based on these results. It is an important study on the glioma including GBM and it is the first study in that area. So the results are important.
Questions and comments:
Q1: In the introduction, it is expected the authors explain about the impact of IDH1, MGMT, and TERT mutation in glioma . Right now the explanation is subminimal and need further expansion.
A1: We greatly appreciate your insightful comments. The introduction was modified accordingly. Pages 2-3.
Q2: The hypothesis and objectives should be clearly stated in the introduction toward the end.
A2: Gliomas are the most prevalent CNS tumors known as a non-curative malignant lesion. Several studies are trying to find prognostic factors in gliomas. Recently, molecular factors have been found to have a critical role in patients with glioma. Apparently, the studies with higher sample sizes besides long-term follow-up are very valuable regarding glioma treatment. In this study, we report our experience in treating gliomas with different molecular profiles, which could be helpful for future studies and therapeutic plans.
Q3: Figure 2 needs more explanation in the results. It is hard to follow up the results in the figure 2.
A3: Your comments have been immensely helpful; thank you so much. It was modified. Page 7.
Q4: Is there any correlation between MGMT, IDH1, and TERT mutation and with tumors in the “Temporal,” Frontal,” Occipital,” Parietal” and “Insular or cerebellum” lobes of the brain?
A4: Your comments have been incredibly valuable. The new analysis was done, and the new results are included in Figures 5 C & D on page 13.
Q5: What could be the future application of this study?
A5: Thank you kindly for the thoughtful and informative feedback. The future direction is added to the discussion on page 18.
Reviewer 2 Report
Comments and Suggestions for Authors
Comments on the manuscript: cancers-2973988: “Genetic prognostic factors in adult glioma; a 10-year experience at a single institution”.
In this manuscript Behrooz et al. analyzed medical records of 186 patients from an institutional cohort, with diagnosis of glioma, to evaluate clinical and molecular features associated with overall survival.
While this subject is very interesting, given the rarity of brain tumors, and the manuscript presents beautiful illustrations, several issues prevent this study to be acceptable in the current form.
Minor problems:
1- The English language needs careful editing, to avoid sentences like “Patients were grouped based on the degree of glioma” (instead of “glioma grade”).
2- Figure 5b is unnecessary, but given its nice aesthetics, I wonder if it is possible to merge the numbers from figure 5a in the anatomic brain on 5b.
Major problems:
1- The authors aim to study diffuse gliomas, but in the Introduction and elsewhere there is a focus in glioblastoma, which is the most frequent type of glioma. I suggest that the authors adopt the general term “diffuse gliomas” when dealing with different glioma grades.
2- The authors did not adopt the 2021 WHO classification for updating the original diagnoses. Even the recommended use of Arabic number in the grading system (grades 2, 3, and 4 instead of II, III, and IV) was not adopted by the authors (PMCID: PMC8328013).
3- Due to the different molecular profiles that characterize pediatric- and adult-type gliomas, it is not usually recommended to study patients from 11 to 81 years old in the same cohort, unless specific alterations in histone H3 (K27 and G34), KIAA1549-BRAF, and so on, are evaluated. Genetic alterations commonly seen in pediatric gliomas were not evaluated in this analysis.
4- There is no description whatsoever of the methods of molecular profiling and protocols for DNA extraction. In addition, there is insufficient description of how each alteration was evaluated. For example: was IDH1 status defined with next generation sequencing? TERT status with Sanger sequencing? How was MGMT methylation defined? Was EGFR amplification evaluated or only the gene mutation?
5- The lack of additional tests (for example, CDKN2A/B and 1p19q status) prevents the adequate characterization of grades 2 and 3 IDH mutant gliomas and this could affect the survival outcomes.
6- The description of treatments is very confusing. The authors need to define patients who received only chemotherapy, only radiation, or chemoradiation. Moreover, were pediatric and adult patients treated with the same drugs?
7- Figure 4 plots are uninformative in the current format. What do the p-values refer to?
8- The authors refer “allele frequencies” when they actually considered binary final results (gene mutant or wild type).
9- Table 2 refer to “EGFR amplification status”, but the categories listed are “mutant” and “wild type”.
Comments on the Quality of English Language
Overall it is a well-written manuscript, but some editing needs to be done.
Author Response
In this manuscript Behrooz et al. analyzed medical records of 186 patients from an institutional cohort, with diagnosis of glioma, to evaluate clinical and molecular features associated with overall survival. While this subject is very interesting, given the rarity of brain tumors, and the manuscript presents beautiful illustrations, several issues prevent this study to be acceptable in the current form.
Questions and comments:
Q1: The English language needs careful editing, to avoid sentences like “Patients were grouped based on the degree of glioma” (instead of “glioma grade”).
A1: Thank you so much. The manuscript was re-edited.
Q2: Figure 5b is unnecessary, but given its nice aesthetics, I wonder if it is possible to merge the numbers from Figure 5a in the anatomic brain on 5b.
A2: I'm truly thankful for your helpful and constructive comments. The numbers were merged.
Q3: The authors aim to study diffuse gliomas, but in the Introduction and elsewhere there is a focus in glioblastoma, which is the most frequent type of glioma. I suggest that the authors adopt the general term “diffuse gliomas” when dealing with different glioma grades.
A3: We greatly appreciate your insightful comments. The introduction was modified accordingly. Pages 2-3.
Q4: The authors did not adopt the 2021 WHO classification for updating the original diagnoses. Even the recommended use of Arabic number in the grading system (grades 2, 3, and 4 instead of II, III, and IV) was not adopted by the authors (PMCID: PMC8328013).
A4: Thank you incredibly for sharing these valuable thoughts and insights. The grading was updated in the whole manuscript, and the reference was added on page 3.
Q5: Due to the different molecular profiles that characterize pediatric- and adult-type gliomas, it is not usually recommended to study patients from 11 to 81 years old in the same cohort, unless specific alterations in histone H3 (K27 and G34), KIAA1549-BRAF, and so on, are evaluated. Genetic alterations commonly seen in pediatric gliomas were not evaluated in this analysis.
A5: Thank you kindly for the thoughtful and informative feedback. Regarding your comment, we excluded patients (11-20 years) and re-did all analyses.
Q6: There is no description whatsoever of the methods of molecular profiling and protocols for DNA extraction. In addition, there is insufficient description of how each alteration was evaluated. For example: was IDH1 status defined with next generation sequencing? TERT status with Sanger sequencing? How was MGMT methylation defined? Was EGFR amplification evaluated or only the gene mutation?
A6: Your comments have been incredibly valuable. Formalin-fixed GBM tumor samples were encompassed with paraffin selected for DNA extracting according to the Reinfenberger et al. MGMT promoter methylation was analyzed and recorded by methylation-specific polymerase chain reaction (PCR) according to the reported data by Mollemann et al. in 2005. The primer sequences used to detect methylated MGMT promoter sequences were “5-GTT TTT AGA ACG TTT TGC GTT TCG AC-3 and 5-CAC CGT CCC GAA AAA AAA CTC CG-3”. The primer sequences used to detect unmethylated MGMT promoter sequences were “5-TGT GTT TTT AGA ATG TTT TGT GTT TTG AT-3 and 5-CTA CCA CCA TCC CAA AAA AAA ACT CCA-3”. IDH1 specific part of exon 4, comprising the R132 mutation hotspot; however, the whole parts of exon 4 for IDH2 were amplified from genomic DNA by polymerase chain reaction (PCR), and the high-resolution melting curve analysis (HRM) was followed by sequence analysis. Previous worthwhile studies reported amplifications of a 122 bp base pairs length fragment spanning IDH1. Based on the HRM guidance on a Light Cycler 480, HRM analysis was performed, and the result was entered into our study database. Real-time quantitative PCR (qPCR) by Light Cycler 480 format recognized the mRNA expression levels of TERT, which was reported before by Arita et al. Moreover, Light Cycler 480 was used in Relative quantification analyses. TERT-specific primers, which are located in exon 5, were used from formalin-fixed paraffin-embedded samples:” GCCTGAGCTGTACTTTGTC” (P0155), and the reverse primer on exon 6: “CGTGTTCTGGGGTTTGATG” (P0156). TERT mRNA expression measurement was incompatible with human total brain RNA. According to the manufacturer's recommendations, twelve microliters of the RNA isolated from 1 mL of CSF were reverse transcribed using the Superscript VILO cDNA synthesis kit (Invitrogen). Samples were then preamplified using the TaqMan PreAmp Master Mix (Applied Biosystems). Briefly, 12.5 μL of the cDNA was added to the Pre-Amp Master Mix together with all the genes of interest and preamplified for 14 cycles, according to the manufacturer’s recommendations. The samples were then diluted 1:10, and TaqMan quantitative reverse transcription PCR was performed on all sam-ples for all the selected genes. The amplification was performed using ABI PRISM 7500 with the following program: 50°C, 2 min; 95°C, 10 min; 40 cycles of 95°C, 15 s, 60°C, 1 min on standard mode. Logarithmic amplifications were interpreted as positive, and relative quantities versus GAPDH/18S were reported for each analyzed sample. Wild-type EGFR primer sequences: “5ʹ-TATGTCCTCATTGC CCTCAACA.” “3ʹ-CTGATGATCTGCAGGTTTTCCA.” EGFRvIII primer sequences: “5ʹ-CTGCTGGCTGCGCTCTG.” “3ʹ-GTGATCTGTCACCACATAATTACCTTTC.” To pre-pare the templates for Sanger sequencing, genomic DNA was amplified using the BigDye Terminator Cycle Sequencing Kit v3.1 with the same primer pair as pyrose-quencing without biotinylating the reverse primer. To prepare the templates for Sanger sequencing, genomic DNA was amplified using the BigDye Terminator Cycle Sequencing Kit v3.1 with the same primer pair as pyrosequencing without biotinylating the reverse primer.
Q7: The lack of additional tests (for example, CDKN2A/B and 1p19q status) prevents the adequate characterization of grades 2 and 3 IDH mutant gliomas and this could affect the survival outcomes.
A7: Thank you for these valuable remarks. This test was one of our limitations, and it is included in the limitation section on page 18.
Q8: The description of treatments is very confusing. The authors need to define patients who received only chemotherapy, radiation, or chemoradiation. Moreover, were pediatric and adult patients treated with the same drugs?
A8: Thank you so much. The pediatrics were excluded, and Table 1 was corrected.
Q9: Figure 4 plots are uninformative in the current format. What do the p-values refer to?
A9: Thank you so much for your valuable comment. For instance, based on Chi-square estimates, the observed ratios deviated from the expected ratios for patients. In grade 4, where radiotherapy was administered, the observed number of surviving patients was 8, while the estimated value was 4.4, and the observed number of deceased patients was 31, whereas the estimated value was 35. This suggests that the application of radiotherapy resulted in an increase in the number of survivors and a decrease in the number of deceased patients. Conversely, in patients who did not receive radiotherapy, the observed number of deceased patients was 55, which was higher than the estimated value of 51.2 by about 4 patients. Additionally, the observed number of surviving patients without radiotherapy was 3, while the estimated value was 6.6. These differences contributed to the significance of the chi-square test.
Q10: The authors refer “allele frequencies” when they actually considered binary final results (gene mutant or wild type).
A10: Your comments have been incredibly valuable. It was corrected.
Q11: Table 2 refer to “EGFR amplification status”, but the categories listed are “mutant” and “wild type”.
A11: It has been a great pleasure to read your comments. It was corrected accordingly.
Reviewer 3 Report
Comments and Suggestions for Authors
The following are my comments about Manuscript ID: cancers-2973988.
Overall, I thought the information was clearly presented and graphic presentation of the data was well done.
Believe that there is a typo on p.13, line 301: the authors indicate that patients with mutant IDH1/2GBM had superior outcomes to those with wild-type IDH tumors, with a survival of 14 and 42 months respectively. The use of respectively indicates that data for the ones with the superior outcomes should be listed first and those with the poorer ones second, in other words it should read a survival of 42 and 14 months respectively.
In the introduction the authors need to provide a strong justification for the biological variables they selected for study.
Is there any indication of why women with IDH1 mutations were concentrated in the group with the best prognosis - could it be hormonal?
The results seem to indicate that even when significant differences in association of a biological variable and survival time was seen that the added time was relatively short, with a maximum of 4 months or so. While interesting, in the discussion the authors should indicate what they think should be done with this information. For example: was there any information on the quality of life the patients enjoyed during the extra time, how might the information be used as a basis for future studies, and are they recommending that these biological variables be routinely monitored and how should the information be stored/made readily available to all? In other words where do they see this work leading. The omission of forward thinking dampness this reviewer”s enthusiasm.
Overall, the paper is well wrtten and with the edits requested could be acceptable.
Author Response
Overall, I thought the information was clearly presented, and the graphic presentation of the data was well done. The paper is well written and, with the edits requested, could be acceptable.
Q1: Believe that there is a typo on p.13, line 301: the authors indicate that patients with mutant IDH1/2GBM had superior outcomes to those with wild-type IDH tumors, with a survival of 14 and 42 months, respectively. The use of respectively indicates that data for the ones with the superior outcomes should be listed first and those with the poorer ones second; in other words, it should read a survival of 42 and 14 months, respectively.
A1: Your insightful comments are greatly appreciated. It was corrected.
Q2: In the introduction, the authors need to provide a strong justification for the biological variables they selected for study.
A2: We greatly appreciate your insightful comments. The introduction was modified accordingly. Pages 2-3.
Q3: Is there any indication of why women with IDH1 mutations were concentrated in the group with the best prognosis - could it be hormonal?
A3: I'm truly thankful for your helpful and constructive comments. The answer might be yes.
Based on our results, women with IDH1 mutations were concentrated in the group with the best prognosis. Sex variations in disease incidence and prognosis are well acknowledged but seldom understood enough to permit sex-specific therapy. Endocrinology and cancer research shows gonadal steroid hormones contribute to GB development and prevalence [PMID: 33752729]. According to some studies, females have a longer survival rate than males [PMID: 15959601, PMID: 16470608]. In an orthotopic model of glioblastoma, Barone et al. [PMID: 19415456] demonstrated that estrogen increased survival, and a study based on estradiol may be beneficial for treating GBM. Observations by Li et al. [ PMID: 9671399] indicate that estrogen protects patients from GBM by methylating estrogen receptors. Yu et al. [PMID: 25315188] also found that androgen receptor signaling promoted GBM tumorigenesis by inhibiting TGF-β (transforming growth factor β) receptor signaling. Another study suggests estrogen may protect against GBM genesis and promote a more favorable biology once GBM occurs [PMID: 30305382]. Pages 16-17
Researchers examined gene expression data from glioblastoma tumors in The Cancer Genome Atlas (TCGA) to determine the biological cause of this sex variation in treatment response. A differentially expressed gene study found that cell division genes strongly influenced male survival. The main strategy for female survival requires integrin gene expression, which spreads malignancies. Individuals with low integrin-signaling component expression survived just over 3 years following diagnosis, compared with little over a year for individuals with other molecular profiles. Men with cancers that expressed minimal cell-cycle signaling components lived more than 18 months, whereas those with other tumor characteristics survived just over a year. The best-survival cluster was all women with IDH1-mutant tumors, whereas males had them in all groups. The data implies that IDH1 mutations may affect GB survival differently by gender [PMID: 30602536]. Pages 16-17
Q4: The results seem to indicate that even when significant differences in association of a biological variable and survival time were seen, the added time was relatively short, with a maximum of 4 months or so. While interesting, in the discussion, the authors should indicate what they think should be done with this information. For example: was there any information on the quality of life the patients enjoyed during the extra time, how might the information be used as a basis for future studies, and are they recommending that these biological variables be routinely monitored, and how should the information be stored/made readily available to all? In other words, where do they see this work leading? The omission of forward thinking dampness this reviewer’s enthusiasm.
A4: We greatly appreciate your insightful comments. As you may know, gliomas are very challenging, and despite new therapeutic approaches, their treatment is still controversial. All efforts to treat these lesions can extend the progression-free survival rate and overall survival rate to around a few months, but overall survival and quality of life are two different topics. Increased OS does not mean increased quality of life. So, in this study, we evaluated prognostic factors and overall survival. Assessing the effect of these factors on QOL will be the title of the other study.
Round 2
Reviewer 2 Report
Comments and Suggestions for Authors
Comments on the reviewed manuscript: cancers-2973988: “Genetic prognostic factors in adult glioma; a 10-year experience at a single institution”.
This is the revised manuscript submitted by Behrooz et al., who analyzed medical records of 186 patients from an institutional cohort, with diagnosis of glioma, to evaluate clinical and molecular features associated with overall survival.
Some issues still prevent this study to be acceptable in the current form.
Major problems:
1- Although the authors make an extensive paragraph explaining the 2021 WHO classification in lines 122-131, the WHO classification is not entirely applied in this manuscript. The term “IDH mutant glioblastoma” is no longer used. Since 2021, based on IDH status, grade 4 diffuse astrocytomas are named “Glioblastoma” (IDH wt astrocytomas of any histological grade with molecular/histological criteria) and “Grade 4 Astrocytoma IDH mutant” – for a summary of criteria, refer to PMID: 34185076. This is specifically an issue in the following exert: “Further analysis revealed that IDH mutations indicate a positive disease prognosis, resulting in longer median survival times in cases of GB (IDH wild type: 15 months; IDH mutant: 31 months) (23). While IDH-mutated glioblastoma often has a more favorable clinical prognosis…” (Lines 97-100).
2- The hypermutation phenotype referred by the authors is more related to use of temozolomide than to IDH mutations themselves – “Furthermore, IDH-mutated glioma is more likely to acquire a hypermutation phenotype, which is linked to a more unfavorable prognosis (25).” (Lines 102-104).
3- Please explain the short form of “TERT-p mutations” and “MGMT-p” (mutations of the promoter region of TERT and MGMT) before their first use in Line 105 and line 164, respectively.
4- Low grade gliomas with TERT-p mutations are usually 1p19q codeleted oligodendrogliomas IDHmutant, which may contribute to better prognosis – “In low-grade gliomas (Grades 2 & 3), TERT-p mutant patients have a better prognosis than wildtype patients, but in GBs (Grade IV), TERT-p mutations are associated with poor outcomes” (lines 107-109). Note also the Roman numerical system in the glioblastoma grade.
5- The authors describe in cohort of 178 patients, 56 treated with radiation plus chemotherapy, 11 with radiation alone, and 5 with exclusive chemotherapy. Was this exclusive chemotherapy done with temozolomide or with other drug(s)? Where all other patients treated with surgery only and expectant conduct? The radiation plus chemotherapy was with temozolomide or other drug(s)? Please expand on the treatment.
6- While the cohort in the current version is smaller than the original (178 versus 186), it is not clear in the text that the authors excluded the pediatric patients. Although Table 2 mentions age groups > or <50 years and range, this information should be in the description of the cohort.
7- “The distribution of MGMT methylation and unmethylation frequencies demonstrated distinctive patterns across different tumor locations within the lobe” – I guess the authors meant “within the brain” (Line 330). In addition, there is no “multilobar region” – It would be better to describe as “multilobar tumors” (Line 332).
8- It is very interesting that most grade 4 gliomas (81.3%) showed IDH1 mutation. This is very different from the usual epidemiology of grade 4 astrocytomas. I wonder if there is any genetic background or environmental factor to explain this finding and encourage the authors to explore this hypothesis in their discussion.
Thank you for the opportunity to review the edited version of this manuscript.
Comments on the Quality of English Language
Minor problems:
1- The English language still needs some editing, to avoid sentences like “The most prevalent central nervous system (CNS) neoplasm arising from glial cells is glioma.” (Abstract). “Glioma” is a generic term, that encompasses 3 different entities: astrocytomas, oligodendrogliomas, and ependymomas, therefore the statement is quite inaccurate. I wonder why the authors did not choose the first sentence of the Introduction (or part of it) for the Abstract. In the Introduction it is written: “Gliomas are primary brain lesions involving cerebral structures without well-defined boundaries and constitute the most prevalent central nervous system (CNS) neoplasms.
2- There are still some instances where the authors used Roman numerals for tumor grade (“In instances of WHO grade IV glioblastoma (GB), IDH mutations are often seen in secondary GB” – line 95), among others (Line 108).
Author Response
#Reviewer 2
This is the revised manuscript submitted by Behrooz et al., who analyzed the medical records of 186 patients from an institutional cohort with glioma diagnoses to evaluate clinical and molecular features associated with overall survival.
Some issues still prevent this study from being acceptable in its current form
Questions and comments:
Q1: Although the authors make an extensive paragraph explaining the 2021 WHO classification in lines 122-131, the WHO classification is not entirely applied in this manuscript. The term “IDH mutant glioblastoma” is no longer used. Since 2021, based on IDH status, grade 4 diffuse astrocytomas are named “Glioblastoma” (IDH wt astrocytomas of any histological grade with molecular/histological criteria) and “Grade 4 Astrocytoma IDH mutant” – for a summary of criteria, refer to PMID: 34185076. This is specifically an issue in the following exert: “Further analysis revealed that IDH mutations indicate a positive disease prognosis, resulting in longer median survival times in cases of GB (IDH wild type: 15 months; IDH mutant: 31 months) (23). While IDH-mutated glioblastoma often has a more favorable clinical prognosis…” (Lines 97-100).
A1: We greatly appreciate your insightful comments. For more consistency in reference [PMID: 34185076, Reference 41 in the manuscript], references 23 and 25 were omitted.
Q2: The hypermutation phenotype referred by the authors is more related to use of temozolomide than to IDH mutations themselves – “Furthermore, IDH-mutated glioma is more likely to acquire a hypermutation phenotype, which is linked to a more unfavorable prognosis (25).” (Lines 102-104).
A2: Your comments have been immensely helpful. We agree with the respected reviewers' comments. We omit this sentence regarding to be consistence reference [PMID: 34185076, Reference 41 in the manuscript].
Q3: Please explain the short form of “TERT-p mutations” and “MGMT-p” (mutations of the promoter region of TERT and MGMT) before their first use in Line 105 and line 164, respectively.
A3: Your comments have been immensely helpful; It was applied in the manuscript accordingly.
Q4: Low grade gliomas with TERT-p mutations are usually 1p19q codeleted oligodendrogliomas IDHmutant, which may contribute to better prognosis – “In low-grade gliomas (Grades 2 & 3), TERT-p mutant patients have a better prognosis than wildtype patients, but in GBs (Grade IV), TERT-p mutations are associated with poor outcomes” (lines 107-109). Note also the Roman numerical system in the glioblastoma grade.
A4: Your comments have been incredibly valuable. It was corrected.
Q5: The authors describe in cohort of 178 patients, 56 treated with radiation plus chemotherapy, 11 with radiation alone, and 5 with exclusive chemotherapy. Was this exclusive chemotherapy done with temozolomide or with other drug(s)? Where all other patients treated with surgery only and expectant conduct? The radiation plus chemotherapy was with temozolomide or other drug(s)? Please expand on the treatment.
A5: Thank you for the thoughtful and informative feedback. The exclusive was based on temozolomide as a chemotherapy drug. It was also added to Figure 1.
Q6: While the cohort in the current version is smaller than the original (178 versus 186), it is not clear in the text that the authors excluded the pediatric patients. Although Table 2 mentions age groups > or <50 years and range, this information should be in the description of the cohort.
A6: Thank you kindly for the thoughtful and informative feedback. It was added to the materials and method on line 139.
Q7: The distribution of MGMT methylation and unmethylation frequencies demonstrated distinctive patterns across different tumor locations within the lobe” – I guess the authors meant “within the brain” (Line 330). In addition, there is no “multilobar region” – It would be better to describe as “multilobar tumors” (Line 332).
A7: Thank you kindly for the thoughtful and informative feedback. It was corrected.
Q8: It is very interesting that most grade 4 gliomas (81.3%) showed IDH1 mutation. This is very different from the usual epidemiology of grade 4 astrocytomas. I wonder if there is any genetic background or environmental factor to explain this finding and encourage the authors to explore this hypothesis in their discussion.
A8: Thank you kindly for the thoughtful and informative feedback. One paragraph was added to address the respected reviewers' comments on page 16.
Q9: The English language still needs some editing, to avoid sentences like “The most prevalent central nervous system (CNS) neoplasm arising from glial cells is glioma.” (Abstract). “Glioma” is a generic term, that encompasses 3 different entities: astrocytomas, oligodendrogliomas, and ependymomas, therefore the statement is quite inaccurate. I wonder why the authors did not choose the first sentence of the Introduction (or part of it) for the Abstract. In the Introduction it is written: “Gliomas are primary brain lesions involving cerebral structures without well-defined boundaries and constitute the most prevalent central nervous system (CNS) neoplasms.
A9: Your comments have been incredibly valuable. The first sentence of the introduction was replaced with the first sentence of the abstract.
Q10: There are still some instances where the authors used Roman numerals for tumor grade (“In instances of WHO grade IV glioblastoma (GB), IDH mutations are often seen in secondary GB” – line 95), among others (Line 108).
A10: We greatly appreciate your insightful comments. It was corrected.